Identification of novel mutations in RNA-dependent RNA polymerases of SARS-CoV-2 and their implications on its protein structure

Chand Gyanendra Bahadur 1
http://orcid.org/0000-0002-7171-1294 Banerjee Atanu 2
http://orcid.org/0000-0001-5478-526X Azad Gajendra Kumar 1 gkazad@patnauniversity.ac.in
1 Department of Zoology, Patna University , Patna, Bihar , India
2 Department of Zoology, Samastipur College , Samastipur, Bihar , India
Gillespie Joseph
Electronic publication date: 2020 Jul 3
Publication date: 2020
Volume: 8
Electronic Location ID: e9492
Received 2020 May 10; Accepted 2020 Jun 16
Copyright: © 2020 Chand et al.
Copyright year: 2020
Copyright holder: Chand et al.
License: This is an open access article distributed under the terms of the Creative Commons Attribution License, which permits unrestricted use, distribution, reproduction and adaptation in any medium and for any purpose provided that it is properly attributed. For attribution, the original author(s), title, publication source (PeerJ) and either DOI or URL of the article must be cited.
License URL: https://creativecommons.org/licenses/by/4.0/

Keywords: COVID-19, SARS-CoV-2, RNA-dependent RNA polymerases (RdRp), Nsp12, Mutation, Indian geographical area

Funding: The authors received no funding for this work.

==============================
The rapid development of the SARS-CoV-2 mediated COVID-19 pandemic has been the cause of significant health concern, highlighting the immediate need for effective antivirals. SARS-CoV-2 is an RNA virus that has an inherently high mutation rate. These mutations drive viral evolution and genome variability, thereby facilitating viruses to have rapid antigenic shifting to evade host immunity and to develop drug resistance. Viral RNA-dependent RNA polymerases (RdRp) perform viral genome duplication and RNA synthesis. Therefore, we compared the available RdRp sequences of SARS-CoV-2 from Indian isolates and the ‘Wuhan wet sea food market virus’ sequence to identify, if any, variation between them. Our data revealed the occurrence of seven mutations in Indian isolates of SARS-CoV-2. The secondary structure prediction analysis of these seven mutations shows that three of them cause alteration in the structure of RdRp. Furthermore, we did protein modelling studies to show that these mutations can potentially alter the stability of the RdRp protein. Therefore, we propose that RdRp mutations in Indian SARS-CoV-2 isolates might have functional consequences that can interfere with RdRp targeting pharmacological agents.

Introduction

The SARS-CoV-2 (a member of Coronaviruses) outbreak occurred in Wuhan, China in December 2019, and it became a pandemic by spreading to almost all countries worldwide. The SARS-CoV-2 causes the COVID-19 disease that has created a global public health problem. As of May 8, 2020, more than 3.9 million confirmed COVID-19 cases were reported worldwide with 0.27 million confirmed deaths. As the virus spreads to new locations, it alters its protein sequence by the introduction of mutations in its genome that help it to survive better in the host (Sackman et al., 2017). The β lymphocytes of the host adaptive immune system eventually identify the specific epitopes of the pathogenic antigen and start producing protective antibodies, which in turn results in agglutination and clearance of the pathogen (Alcami & Koszinowski, 2000; Chaplin, 2010; Jensen & Thomsen, 2012). Being an efficient unique pathogen, a virus often mutates its proteins in a manner that it can still infect the host cells, evading the host immune system. Even when fruitful strategies are discovered and engaged, the high rate of genetic change displayed by viruses frequently leads to drug resistance or vaccine escape (McKeegan, Borges-Walmsley & Walmsley, 2002).

The SARS-CoV-2 has a single stranded RNA genome of approximately 29.8 Kb in length and accommodates 14 ORFs encoding 29 proteins that include four structural proteins: Envelope (E), Membrane (M), Nucleocapsid (N) and Spike (S) protein, 16 non-structural proteins (nsp) and nine accessory proteins (Gordon et al., 2020a; Wu et al., 2020) including the RNA dependent RNA polymerase (RdRp) (also named as nsp12). RdRp is comprised of multiple distinct domains that catalyse RNA-template dependent synthesis of phosphodiester bonds between ribonucleotides. The SARS-CoV-2 RdRp is the prime constituent of the replication/transcription machinery. The structure of the SARS-CoV-2 RdRp has recently been solved (Gao et al., 2020) and show three distinct domains.

For RNA viruses, the RdRp presents an ideal target because of its vital role in RNA synthesis and absence of host homolog. RdRp is therefore a primary target for antiviral inhibitors such as Remdesivir (Gordon et al., 2020b) that is being considered a potential drug for the treatment of COVID-19. Since RNA viruses constantly evolve owing to the rapid rate of mutations in their genome, we decided to analyse the RdRp protein sequence of SARS-CoV-2 from different geographical regions to see if RdRp also mutates. Here, in the present study, we identified and characterised three mutations in the RdRp protein isolated from India against that of the ‘Wuhan wet sea food market’ (Wu et al., 2020) SARS-CoV-2. Altogether, our data strongly suggest at the prevalence of mutations in the genome of SARS-CoV-2 needs to be considered to develop new approaches for targeting this virus.

Methods

Sequence retrieval

We downloaded all SARS-CoV2 sequences from the NCBI virus database as shown in Table 1. As of now, there are 28 SARS-CoV-2 sequences from India have been deposited in this database, out of which, two sequences are not complete; therefore, we retrieved 26 datasets of SARS-CoV-2 (samples are from Indian). As a reference, we downloaded the sequence of SARS-CoV-2 that was first reported genome sequence deposited in the NCBI virus database from the ‘Wuhan wet sea food market area’ from the early days of COVID-19 pandemic (Wu et al., 2020). This virus was formerly called ‘Wuhan seafood market pneumonia virus’ with the accession number YP_009724389.

Table 1 Details of SARS-CoV-2 sequences used in the analysis.

S.No	Accession Number	Authors	Institute/University	
1	YP_009724389	Wu, F., Zhao, S., et al., 2020	Fudan University, Shanghai, China	
2	QHS34545	Yadav, P.D., et al., 2020	NIV, Pashan, Pune, Maharashtra 411021, India	
3	QIA98582	Potdar, V., et al., 2020	NIV, Pashan, Pune, Maharashtra 411021, India	
4	QJC19489	Pandit, R., Shah, T., et al., 2020	GBRC, Gandhinagar, Gujarat 382010, India	
5	QJF11810	Pattabiraman, C., et al., 2020	NIMHANS, Bangalore, Karnataka 560029, India	
6	QJF11822	Pattabiraman, C., et al., 2020	NIMHANS, Bangalore, Karnataka 560029, India	
7	QJF11834	Pattabiraman, C., et al., 2020	NIMHANS, Bangalore, Karnataka 560029, India	
8	QJF11846	Pattabiraman, C., et al., 2020	NIMHANS, Bangalore, Karnataka 560029, India	
9	QJF11858	Pattabiraman, C., et al., 2020	NIMHANS, Bangalore, Karnataka 560029, India	
10	QJF11870	Pattabiraman, C., et al., 2020	NIMHANS, Bangalore, Karnataka 560029, India	
11	QJF11882	Pattabiraman, C., et al., 2020	NIMHANS, Bangalore, Karnataka 560029, India	
12	QJF77844	Muttineni, R., et al., 2020	VRL, Dept. of Zoology, Osmania University, Hyderabad-500007, India	
13	QJF77856	Muttineni, R., et al., 2020	VRL, Dept. of Zoology, Osmania University, Hyderabad-500007, India	
14	QJF77868	Muttineni, R., et al., 2020	VRL, Dept. of Zoology, Osmania University, Hyderabad-500007, India	
15	QJF77880	Muttineni, R., et al., 2020	VRL, Dept. of Zoology, Osmania University, Hyderabad-500007, India	
16	QJQ27840	Pattabiraman, C., et al., 2020	NIMHANS, Bangalore, Karnataka 560029, India	
17	QJQ27852	Pattabiraman, C., et al., 2020	NIMHANS, Bangalore, Karnataka 560029, India	
18	QJQ27864	Pattabiraman, C., et al., 2020	NIMHANS, Bangalore, Karnataka 560029, India	
19	QJQ27876	Pattabiraman, C., et al., 2020	NIMHANS, Bangalore, Karnataka 560029, India	
20	QJQ28343	Trivedi, P., et al., 2020	GBRC, Gandhinagar, Gujarat 382011, India	
21	QJQ28355	Hinsu, A., et al., 2020	GBRC, Gandhinagar, Gujarat 382011, India	
22	QJQ28367	Sabara, P., et al., 2020	GBRC, Gandhinagar, Gujarat 382011, India	
23	QJQ28379	Puvar, A., et al., 2020	GBRC, Gandhinagar, Gujarat 382011, India	
24	QJQ28391	Raval, J., et al., 2020	GBRC, Gandhinagar, Gujarat 382011, India	
25	QJQ28403	Gandhi, M., et al., 2020	GBRC, Gandhinagar, Gujarat 382011, India	
26	QJQ28415	Shah, T., et al., 2020	GBRC, Gandhinagar, Gujarat 382011, India	
27	QJQ28427	Pandya, M., et al., 2020	GBRC, Gandhinagar, Gujarat 382011, India	

Sequence alignments and structure

All the RdRp protein sequences were aligned by multiple sequence alignment platform of CLUSTAL Omega (Madeira et al., 2019). Clustal Omega is a multiple sequence alignment programme that uses seeded guide trees and HMM profile-profile techniques to generate alignments between three or more sequences. The alignment file was carefully studied and differences in the amino acid changes were recorded.

Secondary structure predictions

We used CFSSP (Ashok Kumar, 2013) (Chou and Fasman secondary structure prediction), an online server, to predict secondary structures of SARS-CoV-2 RdRp protein. This server predicts the possibility of secondary structure such as alpha helix, beta sheet, and turns from the amino acid sequence. CFSSP uses Chou-Fasman algorithm, which is based on analyses of the relative frequencies of each amino acid in secondary structures of proteins solved with X-ray crystallography.

RdRp dynamics study

To investigate the effect of mutation on the RdRp protein structural conformation, its molecular stability and flexibility, we used DynaMut software (Rodrigues, Pires & Ascher, 2018) (University of Melbourne, Australia). To run this software we first downloaded the known protein structure of SARS-CoV-2 RdRp from RCSB (PDB ID: 6M71 (Gao et al., 2020), 7BV1 and 7BV2 (Yin et al., 2020)) and used it for analysis. Next, the 6M71, 7BV1 and 7BV2 structure was uploaded on DynaMut software and effect of mutation in various protein structure stability parameters such as vibrational entropy; the atomic fluctuations and deformation energies were determined. DynaMut, a web server implements well established normal mode approaches that is used to analyse and visualise protein dynamics. This software samples conformations and measures the impact of mutations on protein dynamics and stability resulting from vibrational entropy changes and also predicts the impact of a mutation on protein stability.

Results

Identification of mutations in RdRp protein present in Indian isolates

The SARS-COV-2 sequencing data was downloaded from NCBI (NCBI-Virus-SARS-CoV-2 data hub). There are 26 SARS-CoV-2 sequences available from India. We downloaded all Indian SARS-CoV-2 sequences and Wuhan SARS-CoV-2 sequence, which was deposited for the first time after COVID-19 cases started to appear in Wuhan province, China. We downloaded protein sequence of NSP12/RdRp from the database. All these RdRp sequences were first aligned in CLUSTAL Omega to check for similarities or differences. The Clustal Omega algorithm produces a multiple sequence alignment by producing pairwise alignments. Using this algorithm, we identified seven mutations in isolates from Indian SARS-CoV-2 samples compared to Wuhan SARS-CoV-2 RdRp sequence as shown in Fig. 1. We considered the original Wuhan sequence as the wild type for this comparison. These mutations on RdRp are A97V, A185V, I201L, P323L, L329I, A466V and V880I respectively. Out of these, one isolate have triple mutation, two isolates have double mutation, and rest have single mutations (Fig. 1).

Figure 1 Multiple sequence alignment of SARS-CoV-2 RdRp protein.

Multiple sequence alignment of the Wuhan SARS-CoV-2 RdRp protein with sequences obtained from India. The mutations are highlighted in bold/italic font. Only those sequences are shown that have variations; the rest of the sequences are identical among all samples.

P323L mutation causes the alteration in secondary structure of RdRp

Next, we studied the effect of these mutations on secondary structure of RdRp. Our data revealed that mutation at four positions have no effect in secondary structure that includes I201L, L329I, A466V and V880I (Figs. 2C, 2E, 2F and 2G). However, mutation in rest of the three sites causes change in secondary structure (A97V, A185V and P323L) as shown in Figs. 2A, 2B and 2D. At two positions (97 and 185) the alanine amino acid is substituted by valine. The valine side chain is larger than alanine, and substitution of valine at position 97 and 185 impairs packing of the protein as revealed by our secondary structure predications at these two sites (Figs. 2A and 2B, compare i and ii). Our analysis showed that there is addition of two sheet at positions 97 and 98 due to mutation of A97V (Fig. 2A, compare i and ii). Similarly, in A185V mutant, there is a loss of turn at 184 and replacement of helix with sheet structure at 181, 182, 183 and 184 positions (Fig. 2B, compare i and ii). Further, our secondary structure prediction also showed changes in secondary structure when proline is substituted by leucine at 323 position (Figs. 2D and 2H, compare i and ii). The detailed analysis revealed that the mutant RdRp (P323L) have attained considerable changes in secondary structure at the mutation site. There is a loss of turn structures from position 323 and 324 and addition of five sheets at positions 321, 322, 323, 324 and 327 (Figs. 2D and 2H, highlighted in dashed box). Proline possesses a unique property, its side chain cyclizes back on to the backbone amide position, due to which it contributes to the secondary structure formation because of its bulky pyrrolidine ring that places steric constraints on the conformation of the preceding residue in the helix. The substitution of proline to leucine in the mutant RdRp might result in loss of the structural integrity provided by proline. Leucine is a hydrophobic amino acid and generally buried in the folded protein structure. Altogether, the substitution of alanine to valine at position 97, 185 and proline to leucine at position 323 in mutant RdRp is changing the secondary structure of the protein that might have functional consequences.

Figure 2 Prediction of secondary structure of RdRp protein.

Effect of mutations on secondary structure of RdRp. (A–H) demonstrate seven mutations observed in Indian isolates; (i) represents sequence of Wuhan isolate and (ii) represents sequence of Indian isolates. The small rectangular box shows the mutant residue. The difference of secondary structure between Wuhan and Indian isolates are highlighted with position of dashed box in respective panels.

P323L alters the stability dynamics of tertiary structure of RdRp

To understand the impact of mutations on tertiary structure of RdRp, we performed protein modelling using Dynamut software (Rodrigues, Pires & Ascher, 2018). The Dynamut software provide us the information about the alteration in protein stability and flexibility due to the mutations in the native protein structure. Our data revealed that there is change in vibrational entropy energy (ΔΔSVibENCoM) between the wild type (Wuhan isolate) and the mutant Indian isolate (Fig. 3A). Vibrational entropy represents an average of the configurational entropies of the protein within single minima of the energy landscape (Goethe, Fita & Rubi, 2015). The negative ΔΔSVibENCoM of mutant RdRp represents the rigidification of the protein structure and positive ΔΔSVibENCoM represents gain in flexibility. Here, our data show that the mutation at A185V and P323L lead to rigidification of mutant protein structure (Figs. 3B and 3D). However, the mutation at I201L leads to increase in flexibility (Fig. 3C). Further, we also calculated the free energy differences, ΔΔG, between wild-type and mutant. The free energy differences, ΔΔG, caused by mutation have been correlated with the structural changes, such as changes in packing density, cavity volume and accessible surface area and therefore, it measures effect of mutation on protein stability (Eriksson et al., 1992). In general, a ΔΔG below zero means that the mutation causes destabilisation and above zero represents protein stabilisation. Here, our analysis showed positive ΔΔG for A185V and P323L mutations suggesting that P323L mutation is stabilising protein structure (Fig. 3A); however, we observed negative ΔΔG for I201L mutation indicating its destabilising behaviour (Fig. 3A). The ΔΔG for P323L mutant was 0.908 kcal/mol which is significantly higher than others. Therefore, we also did protein modelling of P323L mutant using two additional structures of RdRp (7BV1, 7BV2) (Yin et al., 2020) and analysed its impact on ΔΔG. Our data show that in both the cases (7BV1 and 7BV2) the P323L mutation is leading to stabilisation of the protein structure with the values of ΔΔG as 0.530 kcal/mol and 0.460 kcal/mol respectively.

Figure 3 Effect of mutations on structural dynamics of RdRp protein.

Analysis of RdRp dynamicity and flexibility. (A) The table shows the values of change in ΔΔS ENCoM and ΔΔG due to the mutation. (B, C and D) Δ Vibrational Entropy Energy between Wild-Type and Mutant RdRp, amino acids are coloured according to the vibrational entropy change as a consequence of mutation of RdRp protein. Blue represents a rigidification of the structure and red a gain in flexibility.

We further closely analysed the changes in the intramolecular interactions due to these three mutations in RdRp. Our data showed that it is affecting the interactions of the residues which are present in the close vicinity of alanine, isoleucine and proline. The substitution of wild type residues with mutant residues alters the side chain leading to change of intramolecular bonds in the pocket, where these amino acids resides as shown in Figs. 4A–4C. Therefore, it can be conclusively stated that the three mutations namely, A97V, A185V and P323L, in large, are changing the stability and intramolecular interactions in the protein that might have functional consequences.

Figure 4 Effect of amino acid substitution on interatomic interactions.

Interatomic interactions mediated by A185V, I201L and P323L of RdRp- (A and B) represent alanine to valine substitution at 185th position, (C and D) represent isoleucine to leucine substitution at 201st position, (E and F) represent proline to leucine substitutions at 323rd position. Wild-type and mutant residues are coloured inlight-greenand are also represented as sticks alongside with the surrounding residues which are involved on any type of interactions.

Discussion

RNA viruses including coronaviruses exhibits high mutation rate that provide these viruses evolutionary advantage to better adapt and survivability (Domingo & Holland, 1997). Mutations derive the process of natural selection by selecting those viral strains that are more potent and fit (Duffy, 2018) that facilitates drug resistance and immune evasions (Irwin et al., 2016). The SARS-CoV-2 infected humans from Wuhan province in China quickly spread to almost all countries worldwide. Since, this virus has already spread to different demographic areas having different climatic condition, temperature, humidity and seasonal variations; therefore, we can predict that this virus might be mutating to adapt to new environments. Towards this, we investigated the mutations in RdRp of SARS-CoV-2. We focused primarily on RdRp because it is an indispensable protein that helps in its replication and transcription. More importantly, there are many drugs which specifically target RdRp and are potent antivirals. Our study report the RdRp mutations present in the Indian isolates of SARS-CoV-2 and, will help to understand the effect of variations on RdRp protein. Our data demonstrate that A97V, A185V and P323L mutations lead to significant changes in the protein secondary structure (Fig. 2). P323L mutation lies in the interface domain (residues A250-R365) of the RdRp protein. This domain helps in the coordination of N and C terminal domains of RdRp; therefore, a mutation in the interface domain might have drastic impact on the function of RdRp. A recent virtual molecular docking studies, screened approximately 7,500 drugs to identify SARS-CoV-2 RdRp inhibitor revealed several potential compounds (DOI 10.20944/preprints202003.0024.v1) namely, Simeprevir, Filibuvir and Tegobuvir, etc. Same study also predicted that these drugs bind RdRp at a putative docking site (a hydrophobic cleft) that includes phenylalanine at 326th position. Interestingly, the mutation identified in our study is very close to the docking site (P323L and L329I). Therefore, it is reasonable that the substitution of these two amino acids at 323 and 329 might interfere with the interaction of these drugs with RdRp. Further, our data also revealed that P323L mutation is causing stabilisation of the protein structure. The mutations in RdRp have already been linked to drug resistance in different viruses. Such as a study on RdRp of influenza A virus demonstrate that a K229R mutation confers resistance to favipiravir (Goldhill et al., 2018). Similarly, mutation in hepatitis C virus RdRp at P495, P496 or V499 & T389 and have been linked to drug resistance (Delang et al., 2012). Altogether, it is conceivable that many RNA viruses acquire drug resistance through mutations in RdRp. However, the functional characterisation of RdRp mutations investigated in our study needs to be carried out to understand the exact role of these mutations which will help scientific community to better therapeutic targeting of SARS-CoV-2.

Conclusions

Altogether, our data strongly suggest that SARS-CoV-2 is acquiring mutations as it is spreading to new locations. Most likely, these mutations are helping SARS-CoV-2 to adapt better inside hosts and in new geographical locations. One of the mutations identified in our study (P323L) might have functional consequences that need to be addressed in future studies.

We would like to thank Patna University, Patna, Bihar (India) for providing necessary infrastructural support.

Additional Information and Declarations

Competing Interests

Author Contributions

Data Availability

The authors declare that they have no competing interests.

Gyanendra Bahadur Chand performed the experiments, analysed the data, authored or reviewed drafts of the paper, and approved the final draft.

Atanu Banerjee performed the experiments, analysed the data, prepared figures and/or tables, authored or reviewed drafts of the paper, and approved the final draft.

Gajendra Kumar Azad conceived and designed the experiments, performed the experiments, analysed the data, prepared figures and/or tables, authored or reviewed drafts of the paper, and approved the final draft.

The following information was supplied regarding data availability:

The raw data used in this study are available at the NCBI-Virus-database: YP_009724389, QHS34545, QIA98582, QJC19489, QJF11810, QJF11822, QJF11834, QJF11846, QJF11858, QJF11870, QJF11882, QJF77844, QJF77856, QJF77868, QJF77880, QJQ27840, QJQ27852, QJQ27864, QJQ27876, QJQ28343, QJQ28355, QJQ28367, QJQ28379, QJQ28391, QJQ28403, QJQ28415, QJQ28427.

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
