# Peer review of "Identification of novel mutations in RNA-dependent RNA polymerases of SARS-CoV-2 and their implications on its protein structure"

_PeerJ, doi:10.7717/peerj.9492_

## Round 0.1 · original submission · Minor Revisions

Dear Dr. Chand and colleagues:

Thanks for submitting your manuscript to PeerJ. I have now received three independent reviews of your work, and as you will see, the reviewers raised some minor concerns about the research. Despite this, these reviewers are optimistic about your work and the potential impact it will have on research studying SARS-CoV-2 RNA-dependent RNA polymerases. Thus, I encourage you to revise your manuscript, accordingly, taking into account all of the concerns raised by both reviewers.

Please expedite your revision as this work is timely given the current pandemic and we will kove to except your work as soon as possible.

I look forward to seeing your revision, and thanks again for submitting your work to PeerJ.

Good luck with your revision,

-joe

Reviewer 1 ·

Basic reporting

No comment

Experimental design

No comment

Validity of the findings

No comment

Additional comments

This is a very interesting observation of RdRp mutations in SARS-CoV-2 in a group of viruses isolated from India. Three mutations were predicted to influence secondary structure of RdRp, in particular one of the most common mutations found currently in the GISAID database (P323L). The authors suggest that this P323L mutation will cause resistance to anti-virals. However, since generally viruses trend towards attenuation, could this be an attenuating mutation instead? Could the authors comment on this?

Reviewer 2 ·

Basic reporting

The manuscript by Chand G.B. et al compares the available RdRp sequences of SARS-CoV-2 from Indian isolates and ‘Wuhan wet sea food market virus’ sequences.
The authors report seven mutations observed in Indian SARS-CoV-2 isolates and three unique mutations that showed changes in the secondary structure of the RdRp protein at region of mutation. They also studied molecular dynamics using normal mode analyses and found that these mutations alter the stability of RdRp protein. Based on their analysis, the authors then propose that RdRp mutations in Indian SARS-CoV-2 isolates might have functional consequences that can interfere with RdRp targeting pharmacological agents.

Experimental design

The experimental design is sound. The research question is well defined. The investigation standard are good. Methods are straightforward.

Validity of the findings

The findings are novel, their impact important. Conclusions are well stated.

Additional comments

The authors compares the available RdRp sequences of SARS-CoV-2 from Indian isolates and ‘Wuhan wet sea food market virus’ sequences.
They report seven mutations observed in Indian SARS-CoV-2 isolates and three unique mutations that showed changes in the secondary structure of the RdRp protein at region of mutation. The authors also studied molecular dynamics using normal mode analyses and found that these mutations alter the stability of RdRp protein. Based on their analysis, they then propose that RdRp mutations in Indian SARS-CoV-2 isolates might have functional consequences that can interfere with RdRp targeting pharmacological agents. The manuscript is clearly written, the analysis is well performed and the data are clearly presented. A few typos need to be corrected.

Reviewer 3 ·

Basic reporting

This paper reports mutations in RdRp from SARS2-Cov-2 from Indian isolates.

Experimental design

Based on sequence alignment and structural analysis, the authors identified these mutations in the RdRP and determined their possible effects on the structure integrity of RdRP through the use of molecular dynamic (MD) modeling.

Validity of the findings

The RdRP mutations include A97V, A185V, I201L, P323L, L329I, A466V and V880I. The modeling studies suggest some of mutations can stabilize or destabilize RdRP. In addition, the authors should use the recent structure of RNA-RdRP complex structure to analyze these mutations on the replication ability of RdRP (PDB code 7BV1 and 7BV2 in https://science.sciencemag.org/content/early/2020/04/30/science.abc1560.abstract).

Additional comments

Overall, this is an interesting paper and should be considered for publication ASAP.

---

## Round 0.2 · accepted · Accept

Dear Dr. Chand and colleagues:

Thanks for revising your manuscript based on the concerns raised by the reviewer. I now believe that your manuscript is suitable for publication. Congratulations! I look forward to seeing this work in print, and I anticipate it being an important resource for groups studying Covid-19 and specifically SARS-CoV-2 RNA polymerases. Thanks again for choosing PeerJ to publish such important work.

Best,

-joe